# *Action from Still Image* Dataset and Inverse Optimal Control to Learn Task Specific Visual Scanpaths

**Stefan Mathe**[1,3]   **and**   **Cristian Sminchisescu**[2,1]

[1]Institute of Mathematics of the Romanian Academy of Science
[2]Department of Mathematics, Faculty of Engineering, Lund University
[3]Department of Computer Science, University of Toronto

stefan.mathe@imar.ro, cristian.sminchisescu@math.lth.se

## Abstract

Human eye movements provide a rich source of information into the human visual information processing. The complex interplay between the task and the visual stimulus is believed to determine human eye movements, yet it is not fully understood, making it difficult to develop reliable eye movement prediction systems. Our work makes three contributions towards addressing this problem. First, we complement one of the largest and most challenging static computer vision datasets, VOC 2012 Actions, with human eye movement recordings collected under the primary task constraint of action recognition, as well as, separately, for context recognition, in order to analyze the impact of different tasks. Our dataset is unique among the eyetracking datasets of still images in terms of *large scale* (over 1 million fixations recorded in 9157 images) and different *task controls*. Second, we propose Markov models to automatically discover areas of interest (AOI) and introduce novel sequential consistency metrics based on them. Our methods can automatically determine the *number*, the *spatial support* and the *transitions* between AOIs, in addition to their locations. Based on such encodings, we quantitatively show that given unconstrained read-world stimuli, task instructions have significant influence on the human visual search patterns and are stable across subjects. Finally, we leverage powerful machine learning techniques and computer vision features in order to *learn task-sensitive reward functions* from eye movement data within models that allow to effectively predict the human visual search patterns based on *inverse optimal control*. The methodology achieves state of the art scanpath modeling results.

## 1 Introduction

Eye movements provide a rich source of knowledge into the human visual information processing and result from the complex interplay between the visual stimulus, prior knowledge of the visual world, and the task. This complexity poses a challenge to current models, which often require a complete specification of the cognitive processes and of the way visual input is integrated by them[4, 20]. The advent of modern eyetracking systems, powerful machine learning techniques, and visual features opens up the prospect of learning eye movement models directly from large real human eye movement datasets, collected under task constraints. This trend is still in its infancy, here we aim to advance it on several fronts:

- We introduce a large scale dataset of human eye movements collected under the task constraints of both *action* and *context recognition* from a single image, for the VOC 2012 Actions dataset. The eye movement data is introduced in §3 and is publicly available at http://vision.imar.ro/eyetracking-voc-actions/.

- We present a model to automatically discover areas of interest (AOIs) from eyetracking data, in §4. The model integrates both spatial and sequential eye movement information, in order to better

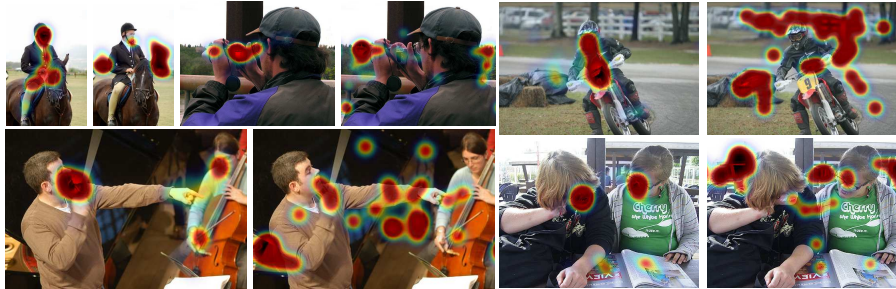

Figure 1: Saliency maps obtained from the gaze patterns of 12 viewers under action recognition (left image in pair) and context recognition (right, in pair), from a single image. Note that human gaze significantly depends on the task (see tab. 1b for quantitative results). The visualization also suggests the existence of stable consistently fixated areas of interest (AOIs). See fig. 2 for illustration.

constrain estimates and to automatically identify the spatial support and the transitions between AOIs in addition to their locations. We use the proposed AOI discovery tools to study inter-subject consistency and show that, on this dataset, task instructions have a significant influence on human visual attention patterns, both spatial and sequential. Our findings are presented in §5.

- We leverage the large amount of collected fixations and saccades in order to develop a novel, fully trainable, eye movement prediction model. The method combines inverse reinforcement learning and advanced computer vision descriptors in order to learn task sensitive reward functions based on human eye movements. The model has the important property of being able to efficiently predict scanpaths of arbitrary length, by integrating information over a long time horizon. This leads to significantly improved estimates. Section §6.2 gives the model and its assessment.

## 2 Related Work

Human gaze pattern annotations have been collected for both static images[11, 13, 14, 12, 26, 18] and for video[19, 23, 15], see [24] for a recent overview. Most of the image datasets available have been collected under free-viewing, and the few task controlled ones[14, 7] have been designed for small scale studies. In contrast, our dataset is both task controlled and more than one order of magnitude larger than the existing image databases. This makes it adequate to using machine learning techniques for saliency modeling and eye movement prediction.

The influence of task on eye movements has been investigated in early human vision studies[25, 3] for picture viewing, but these groundbreaking studies have been fundamentally qualitative. Statistical properties like the saccade amplitude and the fixation duration have been shown to be influenced by the task[5]. A quantitative analysis of task influence on visual search in the context of action recognition from video appears in our prior work[19].

Human visual saliency prediction has received significant interest in computer vision (see [2] for an overview). Recently, the trend has been to learn saliency models from fixation data in images[13, 22] and video[15, 19]. The prediction of eye movements has been less studied. In contrast, predefined visual saliency measures can be used to obtain scanpaths[11] in conjunction with non-maximum suppression. Eye movements have also been modeled explicitly by maximizing the expected future information gain[20, 4] (as one step in [20] or until the goal is reached in [4]). The methods operate on pre-specified reward functions, which limits their applicability. The method we propose shares some resemblance with these later methods, in that we also aim at maximizing the future expected reward, albeit our reward function is learned instead of being pre-specified, and we work in an inverse optimal control setting, which allows, in principle, an arbitrary time horizon. *We are not aware of any eye movement models that are learned from eye movement data.*

## 3 Action from a Single Image – New Human Eye Movement Dataset

One objective of this work is to introduce eye movement recordings for the PASCAL VOC image dataset used for action recognition. Presented in [10], it is one of the largest and most challenging

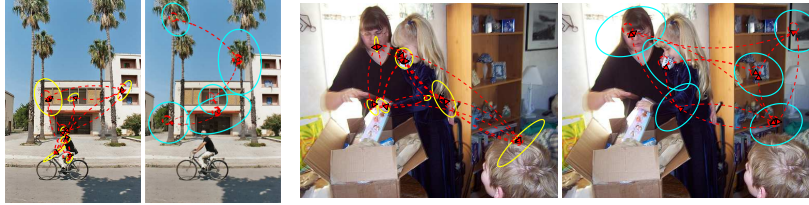

Figure 2: Illustration of areas of interest (AOI) obtained from scanpaths of subjects on three stimuli for the action (left) and context (right) recognition tasks. Ellipses depict states, scaled to match the learned spatial support, whereas dotted arrows illustrate high probability saccades. Visual search patterns are highly consistent both spatially and sequentially and are strongly influenced by task. See fig. 3 and tab. 1 for quantitative results on spatial and sequential consistency.

available datasets of real world actions in static images. It contains 9157 images, covering 10 classes (*jumping*, *phoning*, *playing instrument*, *reading*, *riding bike*, *riding horse*, *running*, *taking photo*, *using computer*, *walking*). Several persons may appear in each image. Multiple actions may be performed by the same person and some instances belong to none of the 10 target classes.

**Human subjects**: We have collected data from 12 volunteers (5 male and 7 female) aged 22 to 46.

**Task**: We split the subjects into two groups based on the given task. The first, action group (8 subjects) was asked to recognize the actions in the image and indicate them from the labels provided by the PASCAL VOC dataset. To assess the effects of task on visual search, we asked the members of the second, context group (4 subjects), to find which of 8 contextual elements occur in the background of each image. Two of these contextual elements – *furniture*, *painting/wallpaper* – are typical of indoors scenes, while the remaining 6 – *body of water*, *building*, *car/truck*, *mountain/hill*, *road*, *tree* – occur mostly outdoors.

**Recording protocol**: The recording setup is identical to the one used in [19]. Before each image was shown, participants were required to fixate a target in the center of a uniform background on the screen. We asked subjects in the action group to solve a multi-target 'detect and classify' task: press a key each time they have identified a person performing an action from the given set and also list the actions they have seen. The exposure time for this task was 3 seconds.[1] Their multiple choice answers were recorded through a set of check-boxes displayed immediately following each image exposure. Participants in the context group underwent a similar protocol, having a slightly lower exposure time of 2.5 seconds. The images were shown to each subject in a different random order.

**Dataset statistics**: The dataset contains 1,085,381 fixations. The average scanpath length is 10.0 for the action subjects and 9.5 for the context subjects, including the initial central fixation. The time elapsed from stimulus display until the first three key presses, averaged over trials in which they occur, are 1, 1.6 and 1.9 seconds, respectively.

## 4 Automatic Discovery of Areas of Interest and Transitions using HMMs

Human fixations tend to cluster on salient regions that generally correspond to objects and object parts (fig. 1). Such areas of interest (AOI) offer an important tool for human visual pattern analysis, *e.g.* in evaluating inter-subject consistency[19] or the prediction quality of different saliency models. Manually specifying AOIs is both time consuming and subjective. In this section, we propose a model to automatically discover the AOI locations, their spatial support and the transitions between them, from human scanpaths recorded for a given image. While this may appear straightforward, we are not aware of a similar model in the literature.

In deriving the model, we aim at four properties. First, we want to be able to exploit not only human fixations, but also constraints from saccades. Consider the case of several human subjects fixating the face of a person and the book she is reading. Based on fixations alone, it can be difficult to separate the book and the person's face into two distinct AOIs due to proximity. Nevertheless, frequent saccades between the book and the person's face provide valuable hints for hypothesizing two distinct, semantically meaningful AOIs. Second, we wish to adapt to an unknown and varying number of AOIs in different images. Third, we want to estimate not only the center of the AOI, but also the spatial support and location uncertainty. Finally, we wish to find the transition probabilities between AOIs. To meet such criteria in a visual representation, we use a statistical model.

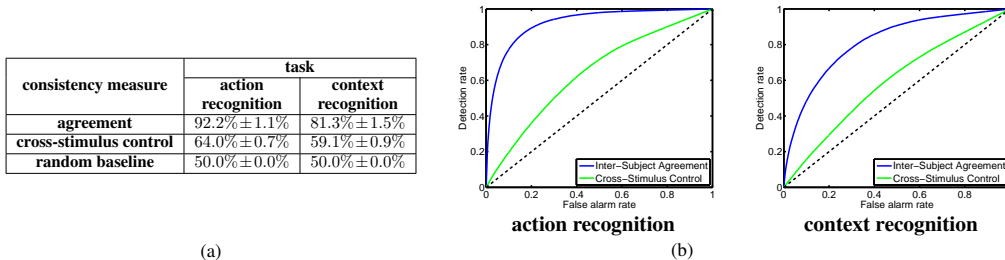

| consistency measure | task | |
|---|---|---|
| | action recognition | context recognition |
| agreement | $92.2\%\pm1.1\%$ | $81.3\%\pm1.5\%$ |
| cross-stimulus control | $64.0\%\pm0.7\%$ | $59.1\%\pm0.9\%$ |
| random baseline | $50.0\%\pm0.0\%$ | $50.0\%\pm0.0\%$ |

(a)

(b)

**action recognition**

**context recognition**

Figure 3: (a) Spatial inter-subject consistency for the tasks of action and context recognition, with standard deviations across subjects. (b) ROC curves for predicting the fixations of one subject from the fixations of the other subjects in the same group on the same image (blue) or on an image (green) randomly selected from the dataset. See tab. 1 for sequential consistency results.

*Image Specific Human Gaze Model*: We model human gaze patterns in an image as a Hidden Markov Model (HMM) where states $\{s_i\}_{i=1}^n$ correspond to AOIs fixated by the subjects and transitions correspond to saccades. The observations are the fixation coordinates $\mathbf{z} = (x, y)$. The emission probability for AOI $i$ is a Gaussian: $p(\mathbf{z}|s_i) = N(\mathbf{z}|\boldsymbol{\mu}_i, \Sigma_i)$, where $\boldsymbol{\mu}_i$ and $\Sigma_i$ model the center and the spatial extent of the area of interest (AOI) $i$. In training, we are given a set of scan-paths $\left\{\boldsymbol{\delta}_j = \left(\mathbf{z}_1, \mathbf{z}_2, \ldots, \mathbf{z}_{t_j}\right)\right\}_{j=1}^k$ and we find the parameters $\boldsymbol{\theta} = \{\boldsymbol{\mu}_i, \Sigma_i\}_{i=1}^n$ that maximize the joint log likelihood $\sum_{j=1}^k \log p(\delta_j|\boldsymbol{\theta})$, using EM[9]. We obtain AOIs, for each image and task, by training the HMM using the recorded human eye scanpaths. We compute the number of states $N^*$ that maximizes the leave-one-out cross validation likelihood over the scanpaths within the training set, with $N \in [1, 10]$. We then re-train the model with $N^*$ states over the entire set of scanpaths.

*Results*: Fig. 2 shows several HMMs trained from the fixations of subjects performing action recognition. On average, the model discovers $8.0$ AOIs for action recognition and $5.6$ for context recognition. The recovered AOIs are task dependent and tend to center on object and object parts with high task relevance, like *phones*, *books*, *hands* or *legs*. Context recognition AOIs generally appear on the background and have larger spatial support, in agreement with the scale of the corresponding structures. There is a small subset of AOIs that is common to both tasks. Most of these AOIs fall on faces, an effect that has also been noted in [6]. Interestingly, some AOI transitions suggest the presence of cognitive routines aimed at establishing relevant relationships between object parts, *e.g.* whether a person is looking at the manipulated object (fig. 2).

The HMM allows us to visualize and analyze the sequential inter-subject consistency (§5) among subjects. It also allows us to evaluate the performance of eye movement prediction models (§6.2).

## 5 Consistency Analysis

Qualitative studies in human vision[25, 16] have advocated a high degree of agreement between the gaze patterns of humans in answering questions regarding static stimuli and have shown that gaze patterns are highly task dependent, although such findings have not yet been confirmed by large-scale quantitative analysis. In this section, we confirm these effects on our large scale dataset for action and context recognition, from a single image. We first study spatial consistency using saliency maps, then analyze sequential consistency in terms of AOI ordering under various metrics.

**Spatial Consistency:** In this section, we evaluate the spatial inter-subject agreement in images.

*Evaluation Protocol*: To measure the inter-subject agreement, we predict the regions fixated by a particular subject from a saliency map derived from the fixations of the other subjects on the same image. Samples represent image pixels and each pixel's score is the empirical saliency map derived from training subjects[14]. Labels are 1 at pixels fixated by the test subject, and 0 elsewhere. For unbiased cross-stimulus control, we check how well a subject's fixations on one stimulus can be predicted from those of the other subjects on a different, unrelated, stimulus. The average precision for predicting fixations on the same stimulus is expected to be much greater than on different stimuli.

*Findings*: Area under the curve (AUC) measured for the two subject groups and the corresponding ROC curves are shown in fig. 3. We find good inter-subject agreement for both tasks, consistent with previously reported results for both images and video [14, 19].

**Sequential Consistency using AOIs**: Next we evaluate the degree to which scanpaths agree in the *order* in which interesting locations are fixated. We do this as a three step process. First, we map each fixation to an AOI obtained with the HMM presented in §4, converting scanpaths to sequences of symbols. Then, we define two metrics for comparing scanpaths, and compute inter-subject agreement in a leave-one-out fashion, for each.

*Matching fixations to AOIs:* We assign a subject's fixation to an AOI, if it falls within an ellipse corresponding to its spatial support (fig. 2). If no match is found, we assign the fixation as *null*. However, due to noise, we allow the spatial support to be increased by a factor. The dashed blue curve in fig. 4c-left shows the fraction (AOIP) of fixations of each human subject, with 2D positions that fall inside AOIs derived from scanpaths of other subjects, as a function of the scale factor. Through the rest of this section, we report results for the threshold to twice the estimated AOI scale, which ensures a 75% fixation match rate across subjects in both task groups.

*AOI based inter-subject consistency:* Once we have converted each scanpath to a sequence of fixations, we define two metrics for inter-subject agreement. Given two sequences of symbols, the *AOI transition* (AOIT) metric is defined as the number of consecutive non-*null* symbol pairs (AOI transitions) that two sequences have in common. The second metric (AOIS), is obtained by *sequence alignment*, as in [19], and represents the longest common subsequence among the two scanpaths. Both metrics are normalized by the length of the longest scanpath. To measure inter-subject agreement, we match the scanpath of each subject $i$ to the scanpaths belonging to other subjects, under the two metrics defined above. The value of the metric for the best match defines the leave-one-out agreement for subject $i$. We then average over all subjects.

*Baselines:* In addition to inter-subject agreement, we define three baselines. First, for cross-stimulus control, we evaluate agreement as in the case of spatial consistency, when the test and reference scanpaths correspond to different randomly selected images. Second, for the random baseline, we generate for each image a set of 100 random scanpaths, where fixations are uniformly distributed across the image. The average metric assigned to these scanpaths with respect to the subjects represents the baseline for sequential inter-subject agreement, in the absence of bias. Third, we randomize the order of each subject's fixations in each image, while keeping their locations fixed, and compute inter-subject agreement with respect to the original scanpaths of the rest of the subjects. The initial central fixation is left unchanged during randomization. This baseline is intended to measure the amount of observed consistency due to the fixation order.

*Findings*: Both metrics reveal considerable inter-subject agreement (table 1), with values significantly higher than for cross-stimulus control and the random baselines. When each subject's fixations are randomized, the fraction of matched saccades (AOIT) drops sharply, suggesting that sequential effects have a significant share in the overall inter-subject agreement. The AOIS metric is less sensitive to these effects, as it allows for gaps in matching AOI sequences.[2]

**Influence of Task**: We will next study the task influence on human visual patterns. We compare the visual patterns of the two subject groups using saliency map and sequential AOI metrics.

*Evaluation Protocol*: For each image, we derive a saliency map from the fixations of subjects doing action recognition, and report the average $p$-statistic at the locations fixated by subjects performing context recognition. We also compute agreement under the AOI-based metrics between the scanpaths of subjects performing context recognition, and subjects from the action recognition group.

*Findings*: Only 44.1% of fixations made during context recognition fall onto action recognition AOIs, with an average $p$-value of 0.28 with respect to the action recognition fixation distribution. Only 10% of the context recognition saccades have also been made by active subjects, and the AOIS metric between context and active subjects' scanpaths is 23.8%. This indicates significant differences between the subject groups in terms of their visual search patterns.

## 6   Task-Specific Human Gaze Prediction

In this section, we show that it is possible to effectively predict task-specific human gaze patterns, both spatially and sequentially. To achieve this, we combine the large amounts of information available in our dataset with state-of-the art visual features and machine learning techniques.

| consistency measure | task | | | | | |
|---|---|---|---|---|---|---|
| | action recognition | | | context recognition | | |
| | **AOIP** | **AOIT** | **AOIS** | **AOIP** | **AOIT** | **AOIS** |
| agreement | 79.9%±1.9% | 34.0%±1.3% | 39.9%±1.0% | 76.4%±2.6% | 35.6%±0.9% | 44.9%±0.4% |
| agreement (random order) | 79.9%±1.9% | 21.8%±0.7% | 31.0%±0.7% | 76.4%±2.6% | 23.2%±0.3% | 35.5%±0.3% |
| cross-stimulus control | 29.4%±0.8% | 4.9%±0.3% | 13.9%±0.3% | 40.0%±2.1% | 7.9%±0.5% | 19.6%±0.2% |
| random scanpaths | 15.5%±0.1% | 1.5%±0.0% | 2.5%±0.0% | 31.9%±0.1% | 4.2%±0.0% | 7.6%±0.0% |

Table 1: Sequential inter-subject consistency measured using AOIs (fig. 2), for both task groups. A large fraction of each subject's fixations falls onto AOIs derived from the scanpaths of the other subjects (AOIP). Significant inter-subject consistency exists in terms of AOI transitions (AOIT) and scanpath alignment score (AOIS).

## 6.1 Task-Specific Human Visual Saliency Prediction

We first study the prediction of human visual saliency maps. Human fixations typically fall onto image regions that are meaningful for the visual task (fig. 2). These regions often contain objects and object parts that have similar identities and configurations for each semantic class involved, *e.g.* the configuration of the legs while running. We exploit this repeatability and represent each human fixation by HoG descriptors[8]. We then train a sliding window detector with human fixations and compare it with competitive approaches reported in the literature.

*Evaluation Protocol*: For each subject group, we obtain positive examples from fixated locations across the training portion of the dataset. Negative examples are extracted similarly at random image locations positioned at least $3^o$ away from all human fixations. We extract 7 HoG descriptors with different grid configurations and concatenate them, then represent the resulting descriptor using an explicit, approximate $\chi^2$ kernel embedding[17]. We train a linear SVM to obtain a detector, which we run in sliding window fashion over the test set in order to predict saliency maps. We evaluate the detector under the AUC metric and the spatial KL divergence criterion presented in [19]. We use three baselines for comparison. The first two are the uniform saliency map and the central bias map (with intensity inversely proportional to distance from center). As an upper bound on performance, we also compute saliency maps derived from the fixations recorded from subjects. The KL divergence score for this baseline is derived by splitting the human subjects into two groups and computing the KL divergence between the saliency maps derived from these two groups, while the AUC metric is computed in a leave-one-out fashion, as for spatial consistency. We compare the model with two state of the art predictors. The first is the bottom-up saliency model of Itti&Koch[11]. The second is a learned saliency predictor introduced by Judd *et al.*[13], which integrates low and mid-level features with several high-level object detectors such as cars and people and is capable to optimally weight these features given a training set of human fixations. Note that many of these objects often occur in the VOC 2012 actions dataset.

*Findings*: Itti&Koch's model is not designed to predict task-specific saliency and cannot handle task influences on visual attention (fig. 4). Judd's model can adapt to some extent by adjusting feature weights, which were trained on our dataset. Out of the evaluated models, we find that the task-specific HoG detector performs best under both metrics, especially under the spatial KL divergence, which is relevant for computer vision applications[19]. Its flexibility stems from its large scale training using human fixations, the usage of general-purpose computer vision features (as opposed, *e.g.*, to the specific object detectors used by Judd *et al.*[13]), and in part from the use of a powerful nonlinear kernel for which good linear approximations are available[17, 1].

## 6.2 Scanpath Prediction via Maximum Entropy Inverse Reinforcement Learning

We now consider the problem of eye movement prediction under specific task constraints. Models of human visual saliency can be used to generate scanpaths, *e.g.* [11]. However, current models are designed to predict saliency for the free-viewing condition and do not capture the focus induced by the cognitive task. Others [20, 4] hypothesize that the reward driving eye movements is the expected future information gain.

Here we take a markedly different approach. Instead of specifying the reward function, we learn it directly from large amounts of human eye movement data, by exploiting policies that operate over long time horizons. We cast the problem as Inverse Reinforcement Learning (IRL), where we aim to recover the intrinsic reward function that induces, with high probability, the scanpaths recorded from human subjects solving a specific visual recognition task. Our learned model can imitate

| baselines | | | | |
|---|---|---|---|---|
| **feature** | **action recognition** | | **context recognition** | |
| | **KL** | **AUC** | **KL** | **AUC** |
| uniform baseline | 12.00 | 0.500 | 11.02 | 0.500 |
| central bias | 9.59 | 0.780 | 8.82 | 0.685 |
| human | 6.14 | 0.922 | 5.90 | 0.813 |
| **predictors** | | | | |
| HOG detector* | **8.54** | **0.736** | **8.10** | **0.646** |
| Itti & Koch[11] | 16.53 | 0.533 | 15.04 | 0.512 |
| Judd *et al.*[13]* | 11.00 | 0.715 | 9.66 | 0.636 |

(a) human visual saliency prediction

| baselines | | | | | | |
|---|---|---|---|---|---|---|
| **feature** | **action recognition** | | | **context recognition** | | |
| | **AOIP** | **AOIT** | **AOIS** | **AOIP** | **AOIT** | **AOIS** |
| human scanpaths | 79.9% | 34.0% | 39.9% | 76.4% | 35.6% | 44.9% |
| random scanpaths | 15.5% | 1.5% | 2.5% | 31.9% | 4.2% | 7.6% |
| **predictors** | | | | | | |
| IRL* | **35.6%** | **6.6%** | **18.4%** | **44.9%** | **11.6%** | **25.7%** |
| Renninger [20] | 24.4% | 2.0% | 14.6% | 40.3% | 7.0% | 23.9% |
| Itti & Koch [11] | 28.6% | 2.7% | 16.8% | 42.9% | 7.5% | 24.1% |

(b) eye movement prediction

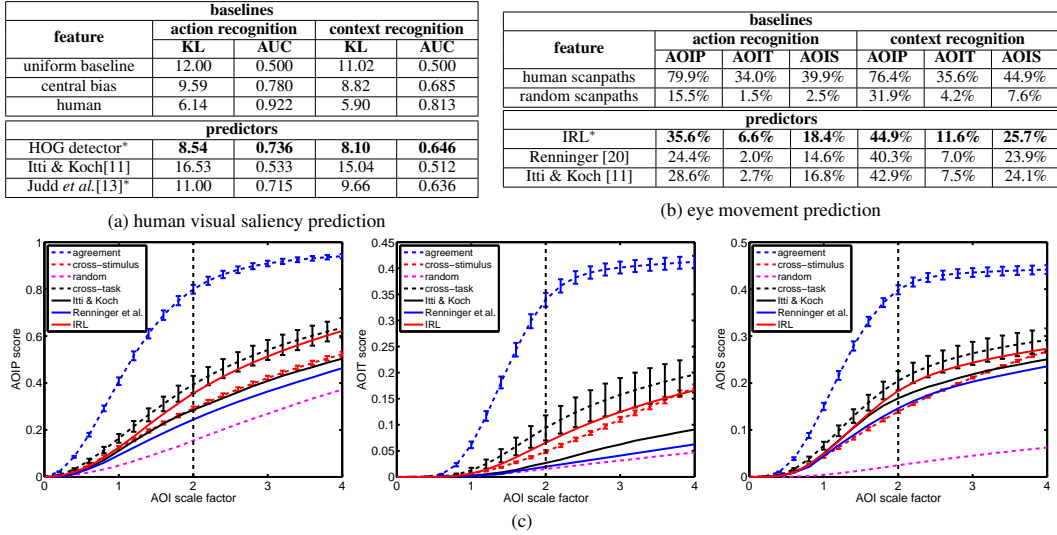

(c)

Figure 4: Task-specific human gaze prediction performance on the VOC 2012 actions dataset. (a) Our trained HOG detector outperforms existing saliency models, when evaluated under both the KL divergence and AUC metrics. (b-c) Learning techniques can also be used to predict eye movements under task constraints. Our proposed Inverse Reinforcement Learning (IRL) model better matches observed human visual search scanpaths when compared with two existing methods, under each of the AOI based metrics we introduce. Methods marked by '*' have been trained on our dataset.

useful saccadic strategies associated with cognitive processes involved in complex tasks such as action recognition, but avoids the difficulty of explicitly specifying these processes.

*Problem Formulation*: We model a scanpath $\boldsymbol{\delta}$ as a sequence of states $\mathbf{s}_t = (x_t, y_t)$ and actions $\mathbf{a}_t = (\Delta x, \Delta y)$, where states correspond to fixations, represented by their visual angular coordinates with respect to the center of the screen, and actions model saccades, represented as displacement vectors expressed in visual degrees. We rely on a maximum entropy IRL formulation[27] to model the distribution over the set $\Delta^{(\mathbf{s},T)}$ of all possible scanpaths of length $T$ starting from state $\mathbf{s}$ for a given image as:

$$p_{\boldsymbol{\theta}}^{(\mathbf{s},T)}(\boldsymbol{\delta}) = \frac{1}{Z^{(T)}(\mathbf{s})} \cdot \exp\left[\sum_{t=1}^{T} r_{\boldsymbol{\theta}}(\mathbf{s}_t, \mathbf{a}_t)\right], \quad \forall \boldsymbol{\delta} \in \Delta^{(\mathbf{s},T)} \tag{1}$$

where $r_{\boldsymbol{\theta}}(\mathbf{s}_t, \mathbf{a}_t)$ is the reward function associated with taking the saccadic action $\mathbf{a}_t$ while fixating at position $\mathbf{s}_t$, $\boldsymbol{\theta}$ are the model parameters and $Z^{(T)}(\mathbf{s})$ is the partition function for paths of length $T$ starting with state $\mathbf{s}$, see (3). The reward function $r_{\boldsymbol{\theta}}(\mathbf{s}_t, \mathbf{a}_t) = \mathbf{f}^\top(\mathbf{s}_t)\boldsymbol{\theta}_{\mathbf{a}_t}$ is the inner product between a feature vector $\mathbf{f}(\mathbf{s}_t)$ extracted at image location $\mathbf{s}_t$ and a vector of weights corresponding to action $\mathbf{a}_t$. Note that reward functions in our formulation depend on the subject's action. This enables the model to encode saccadic preferences conditioned on the current observation, in addition to planning future actions by maximizing the cumulative reward along the entire scanpath, as implied by (1).

In our formulation, the goal of Maximum Entropy IRL is to find the weights $\boldsymbol{\theta}$ that maximize the likelihood of the demonstrated scanpaths across all the images in the dataset. For a single image and given the set of human scanpaths $E$, all starting at the image center $\mathbf{s}^c$, the likelihood is:

$$\mathcal{L}_{\boldsymbol{\theta}} = \frac{1}{|E|} \sum_{\boldsymbol{\delta} \in E} \log p_{\boldsymbol{\theta}}^{(\mathbf{s}^c,T)}(\boldsymbol{\delta}) \tag{2}$$

This maximization problem can be solved using a two step dynamic programming formulation. In the backward step, we compute the state and state-action partition functions for each possible state $\mathbf{s}$ and action $\mathbf{a}$, and for each scanpath length $i = \overline{1, T}$:

$$Z_{\boldsymbol{\theta}}^{(i)}(\mathbf{s}) = \sum_{\boldsymbol{\delta} \in \Delta^{(\mathbf{s},i)}} \exp\left[\sum_{t=1}^{i} r_{\boldsymbol{\theta}}(\mathbf{s}_t, \mathbf{a}_t)\right], \quad Z_{\boldsymbol{\theta}}^{(i)}(\mathbf{s}, \mathbf{a}) = \sum_{\substack{\boldsymbol{\delta} \in \Delta^{(\mathbf{s},i)} \\ \text{s.t.} \\ \mathbf{a}_1 = \mathbf{a}}} \exp\left[\sum_{t=1}^{i} r_{\boldsymbol{\theta}}(\mathbf{s}_t, \mathbf{a}_t)\right] \tag{3}$$

The optimal policy $\pi_{\boldsymbol{\theta}}^{(i)}$ at the $i^{\text{th}}$ fixation is:

$$\pi_{\boldsymbol{\theta}}^{(i)}(\mathbf{a}|\mathbf{s}) = Z_{\boldsymbol{\theta}}^{(T-i+1)}(\mathbf{s},\mathbf{a})/Z_{\boldsymbol{\theta}}^{(T-i+1)}(\mathbf{s}) \qquad (4)$$

This policy induces the maximum entropy distribution $p_{\boldsymbol{\theta}}^{(\mathbf{s}^c,T)}$ over scanpaths for the image and is used in the forward step to efficiently compute the expected mean feature count for each action $\mathbf{a}$, which is $\hat{\mathbf{f}}_{\boldsymbol{\theta}}^{\mathbf{a}} = \mathbb{E}_{\boldsymbol{\delta} \sim p_{\boldsymbol{\theta}}^{(\mathbf{s}^c,T)}} \left[ \sum_{t=1}^{T} \mathbf{f}(\mathbf{s}_t) \cdot \mathbb{I}[\mathbf{a}_t = \mathbf{a}] \right]$, where $\mathbb{I}[\cdot]$ is the indicator function. The gradient of the likelihood function (2) with respect to the parameters $\boldsymbol{\theta}_{\mathbf{a}}$ is:

$$\frac{\partial \mathcal{L}_{\boldsymbol{\theta}}}{\partial \boldsymbol{\theta}_{\mathbf{a}}} = \tilde{\mathbf{f}}^{\mathbf{a}} - \hat{\mathbf{f}}_{\boldsymbol{\theta}}^{\mathbf{a}} \qquad (5)$$

where $\tilde{\mathbf{f}}^{\mathbf{a}} = \frac{1}{|E|} \sum_{\boldsymbol{\delta} \in E} \sum_t \mathbf{f}(\mathbf{s}_t) \cdot \mathbb{I}[\mathbf{a}_t = \mathbf{a}]$ is the empirical feature count along training scanpaths.

Eqs. (1)–(5) are defined for a given input image. The likelihood and its gradient over the training set are obtained by summing up the corresponding quantities. In our formulation policies encode the *image specific* strategy of the observer, based on a *task specific* reward function that is learned across all images. We thus learn two different IRL models, for action and context analysis. Note that we restrict ourselves to scanpaths of length $T$ starting from the center of the screen and do not predefine goal states. We validate $T$ to the average scanpath length in the dataset.

*Experimental Procedure*: We use a fine grid with $0.25^{\text{o}}$ stepsize for the state space. The space of all possible saccades on this grid is too large to be practical ($\approx 10^5$). We obtain a reduced vocabulary of $1,000$ actions by clustering saccades in the training set, using k-means. We then encode all scanpaths in this discrete (state,action) space, with an average positional error of $0.47^{\text{o}}$. We extract HoG features at each grid point and augment them with the output of our saliency detector. We optimize the weight vector $\boldsymbol{\theta}$ in the IRL framework and use a BFGS solver for fast convergence.

*Findings*: A trained MaxEnt IRL eye movement predictor performs better than the bottom up models of Itti&Koch[11] and Renninger *et al.*[20] (fig. 4bc). The model is particularly powerful for predicting saccades (see the AOIT metric), as it can match more than twice the number of AOI transitions generated by bottom up models for the action recognition task. It also outperforms the other models under the AOIP and AOIS metrics. Note that the latter only captures the overall ranking among AOIs as defined by the order in which these are fixated. A gap still remains to human performance, underlining the difficulty of predicting eye movements in real world images and for complex tasks such as action recognition. For context recognition, prediction scores are generally closer to the human baseline. This is, at least in part, facilitated by the often larger size of background structures as compared to the humans or the manipulated objects involved in actions (fig. 2).

## 7  Conclusions

We have collected a large set of eye movement recordings for VOC 2012 Actions, one of the most challenging datasets for action recognition in still images. Our data is obtained under the task constraints of action and context recognition and is made publicly available. We have leveraged this large amount of data (1 million human fixations) in order to develop Hidden Markov Models that allow us to determine fixated AOI locations, their spatial support and the transitions between them automatically from eyetracking data. This technique has made possible to develop novel evaluation metrics and to perform quantitative analysis regarding inter-subject consistency and the influence of task on eye movements. The results reveal that given real world unconstrained image stimuli, the task has a significant influence on the observed eye movements both spatially and sequentially. At the same time such patterns are stable across subjects.

We have also introduced a novel eye movement prediction model that combines state-of-the-art reinforcement learning techniques with advanced computer vision operators to learn task-specific human visual search patterns. *To our knowledge, the method is the first to learn eye movement models from human eyetracking data*. When measured under various evaluation metrics, the model shows superior performance to existing bottom-up eye movement predictors. To close the human performance gap, better image features, and more complex joint state and action spaces, within reinforcement learning schemes, will be explored in future work.

**Acknowledgments:** Work supported in part by CNCS-UEFISCDI under CT-ERC-2012-1.

## Footnotes

[1]Protocol may result in multiple keypresses per image. Exposure times were set empirically in a pilot study.

[2]Although harder to interpret numerically, the negative log likelihood of scanpaths under HMMs also defines a valid sequential consistency measure. We observe the following values for the action recognition task: agreement 9.2, agreement (random order) 13.1, cross-stimulus control 25.8, random baseline 46.6.

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
