[Reviews · NeurIPS 2013]

Submitted by Assigned_Reviewer_5

This paper makes several contributions along the lines of capturing human eye movements. Firstly, they propose a new large-scale dataset containing more than 1 million eyetracking annotations on the PASCAL VOC 2012 Action dataset in a task dependent manner. In addition, they present models for the automatic discovery of areas of interest and infer spatial support from images. These areas of interest are used for consistency analysis between the different annotators, showing a clear effect of the task-specific instructions provided. Furthermore, the authors propose a novel prediction model for dynamic eye movements based on inverse reinforcement learning. The proposed model leverages the large amount of training data and outperforms existing state-of-the-art approaches significantly.

Overall, the paper is well written and deals with an increasingly relevant problem of tracking eye movements and applying them to a variety of tasks. The dataset provided is extremely large compared to existing datasets and will provide a useful benchmark/tool for the community. In addition the proposed experiments clearly validate the claim that the eye movements are task dependent.

It would be interesting to extend this work for action recognition in a similar setting as [Z1] for example but of course it is not possible given the space constraints.

[Z1] Fine-Grained Crowdsourcing for Fine-Grained Recognition, CVPR 2011
Summary: The paper is well written and addresses an increasingly relevant problem in a strong way. The dataset will become an important benchmark in the future. In addition, thorough experimental evaluation is done with some new algorithmic proposals showing promising results.

Submitted by Assigned_Reviewer_6

The paper aims to predict fixations and saccadic motions of the gaze for human subjects observing still images. A new dataset of eye movements is collected for VOC 2012 Actions images. Two separate sets of eye movements are collected from two groups of subjects given two tasks: (a) recognizing the action and (b) recognizing objects in the background. The consistency of subjects in fixations and eye movements is analyzed, by modeling the gaze as transitions between Gaussian states (Areas of Interest). The prediction of gaze locations is done by training an SVM on HOG features sampled at Areas of Interest for positives and at other random locations for negatives.

The paper is clearly written, it has a good overview of the prior art and a thorough description of experimental setups. The proposed methods for learning gaze prediction are somewhat simple but intuitive. Experimental results on gaze prediction are compared to several relevant methods and show improvements. The new dataset and the proposed baselines could be interesting both in cognitive science and for action recognition.

Comments:
- The parameters of the method [12] have been tuned for images of outdoor scenes. Re-estimating parameters of [12] on VOC Actions image and gaze data may improve results of [12] in Table1(a).
- l.207 The computation of ROC curve should be explained: what are the samples, their scores and labels?
- l.249 "The dotted blue curve in fig. 2": I do not see dotted blue curves in fig. 2 and cannot interpret well the paragraph l.246-251
- l.406 "Itti&Koch[17]" -> "Itti&Koch [9]".
- l.286-290: The explanation of different conclusions to that in [16] does not appeal to me. I do not think that VOC action images from Flickr are more "uncontrolled" than movie shots. Maybe yes, but then this claim should be verified. I am not an expert in gaze, but the difference to [16] seems to come from the task. While subjects in this paper were asked to focus on the background, [16] examines gaze under free video viewing which may naturally lead to the focus on actions. I would thus re-formulate the current interpretation of results w.r.t. [16].
Summary: Summary: The paper does a good job of collecting and analyzing static/dynamic gaze data as well as presenting baselines for task-dependent gaze prediction. As I am not an expert in gaze analysis, it's hard for me to tell the relevance of this paper for NIPS. It might be more suited for a cognitive science conference.

Submitted by Assigned_Reviewer_8

This paper presents work on human eye movements when viewing still images. A new dataset is collected, measuring eye fixations under task-specific instructions -- the same images are viewed by subjects instructed to determine the human action or the scene context. The paper presents analyses of these data, a model for scanpaths, a model for fixations, and a temporal model of saccades. The analyses measure consistency of different subjects' viewing of an image-task pair. The model for scanpaths builds a task-image pair-specific HMM. The model for fixations uses HOG descriptors and an SVM to predict salient regions of an image. The temporal model uses reinforcement learning to generate a model for viewing of an image.

Overall, the paper makes contributions to eye movement analysis -- expanding on previous work [16] to clearly show task matters for gaze. In my view this is the strongest part of the paper. On the other hand, various computational models are described, the HMM for scanpaths, the HOG-SVM saliency predictor, and the reinforcement learning. These are reasonable, but there are concerns about their applicability/importance (details below). On the whole, the paper is solid, and could be accepted as a poster.

1) The HMM for scanpaths seems quite limited in its use. The model is a task-image pair-specific HMM -- given an image for which many subjects' eye movements have been observed given a task, an HMM that predicts how other people will view this image when asked to do this task is learned. While HMMs are a reasonable model, this task doesn't seem very common. Further, it seems any temporal model would be appropriate for this task, and no particular baselines are presented.

2) A HOG-SVM saliency predictor is defined. This uses dataset and task-specific training data to build a model of where a subject will look. The performance is good, and better than baselines. However, it seems only marginally better than [12], and this could either be due to the use of different features or a non-linear classifier. It is necessary to clarify from which of these the performance gain comes.

3) A similar comment applies to the reinforcement learning and comparison to baselines. Different features (HOG) and learning strategy are used in this method. While overall performance is important, understanding where these gains come from is important too.


Minor comments:

- The paper needs proofreading, there are many typos and spelling errors.

- Why does the dataset have 3s exposures for action recognition and 2.5 for context? Won't this difference bias comparative studies?

- What are the "three key presses" (line 135)? From the description, it seems only one key press is made by a subject.

- Line 200 "locating people"?

- +/- std. deviation should be added to numbers in Table 1.

- I don't think the cross-stimulus baseline for gaze prediction is defined in the text. A sentence could be added, especially pointing out that the performance is significantly lower, which bolsters a main message of the paper.




Summary: A new dataset of task-specific eye movements for image viewing is presented. A few reasonable computational models are built from these data.
Author Feedback

Author rebuttal: We thank all reviewers for valuable feedback, which we will aim to fully integrate.

== AR5

Extension for action recognition: Although not the objective of this work, we consider the problem of action recognition based on eye movement predictions, as an interesting future direction. Encouraging results along this line have been reported for video[16].

== AR6

Re-estimating the parameters of [12]: This is exactly what we do – we re-estimate the weights of the model in [12] using the training portion of our dataset. Results in table 2 are obtained using those fitted parameters. In the submission, we have marked all methods that have been fitted to our training set by ‘*’-- including [12].

ROC computation: Samples represent image pixels. Each pixel's score is the empirical saliency map derived from training subjects (stdev. 1.5 visual degrees). Labels are 1 at pixels fixated by the test subject, and 0 elsewhere.

Dotted blue curve fig. 2: We apologize for this typo: line 249 should read “tab. 2” instead of “fig. 2”. The dotted blue curve in table 2 shows the fraction of fixations of each human subject that are mapped to AOIs derived from the other subjects. Fig. 2 illustrates the AOIs.

Explanation of conclusions in [16]: We agree: visual behavior is affected both by stimulus type and by task. In [16], both tasks - free viewing and action recognition - are likely to bias attention on foreground, while in our work, context recognition is likely to do the opposite. As available data in [16]does not allow unequivocally identifying the relative influence of these two variables on the measured scanpaths, we will qualify this aspect in the text.

== AR8

HMMs for dynamic visual consistency: The goal of this model is to allow to rigorously and automatically quantify the spatio-temporal structure of scanpaths captured under task-specific constraints. It is not designed as a model for a high-end application. Our HMM formulation jointly, in a single model, can automatically locate AOIs and determine their spatial extent, as well as the temporal ordering induced by task. It also allows for intuitive visual interpretation. Notice that we have a spatio-temporal problem and not only a temporal one. We are not aware of other baselines in the literature, for this problem.

Saliency prediction: The HOG-SVM saliency predictor is significantly better than [12] in terms of the far more relevant for computer vision spatial KL divergence and superior under the AUC metric as well, a trend consistent with results in video [16], where different features are used. This suggests that SVM classifiers with non-linear kernels are well suited for approximating saliency maps as spatial probability distributions.

IRL comparison to baselines: Our model is substantially different from existing baselines: (1) it is fully learned, (2) can incorporate significant lookahead which enables it to form long term spatio-temporal strategies and (3) incorporates stronger local features than other models we compare with (e.g. [17] uses a single histogram of edge orientations).

Exposure times: These have been determined empirically in a pilot study. We wished to allow enough exposure for task completion but limit free viewing. We found that different tasks require different exposure times. We consider both viewing time and the various consistency metrics we introduce, as dependent variables of the task, rather than dependent on each other - e.g. both AOI match scores and the time needed to explore AOIs are affected by their spatial layout, which is task specific.

Key presses: Images from this dataset may contain multiple persons performing actions. We asked subjects to solve a multi-target “detect and classify” task: identify all actions, from the specified list, and press a key each time they have identified a person performing one of them. Thus, there may be multiple keypresses for the same image.

Standard deviations, table 1: We will add standard deviations across subjects to the final version. For the action recognition and inter-subject agreement metric, we have the following values: AOI match = 1.9%, AOI transitions = 1.3%, AOI alignment = 1.0%.

Cross-stimulus baseline: We will include a definition of the evaluation procedure to sec. 5, together with an emphasis on the observed performance gap.

== AR5, AR6, AR8 on Quality score

Respectfully, we do not consider our work to be incremental, or unlikely to make impact, for the following reasons:

- we introduce a database that is not only one order of magnitude larger than existing ones, but unique in being captured under high level action recognition in still images task constraints.

- we are not aware of any work that uses IRL to learn task-specific eye movement models, neither in the image nor in the video domain. Nor are we aware of automatic spatial-temporal consistency models like the ones we propose.

- analyzing and modeling human eye movements from large training sets and based on powerful machine learning tools seems to be a relevant information processing component for both brains and machines – a main focus of NIPS. Studies like [16] and others show that automatic systems trained with human eye movements can achieve state of the art computer vision performance.

- Industrial work in first person vision projects, Google Glass, or wearable computing, seems to critically depend, also, on a successful eye movement modeling component.